# Has Anything Changed in the Frequency of Emergency Department Visits and the Profile of the Adolescent Seeking Emergency Mental Care during the COVID-19 Pandemic?

**DOI:** 10.3390/children10030578

**Published:** 2023-03-17

**Authors:** Valentina Ferro, Roberto Averna, Manuel Murciano, Umberto Raucci, Sebastian Cristaldi, Anna Maria Caterina Musolino, Maria Pontillo, Nicoletta Della Vecchia, Milena Labonia, Mara Pisani, Maria Laura Pucciarini, Raffaella Nacca, Antonino Reale, Stefano Vicari, Alberto Villani, Massimiliano Raponi

**Affiliations:** 1Department of Emergency, Acceptance and General Pediatrics, Bambino Gesù Children’s Hospital, IRCCS, 00165 Rome, Italy; 2Child and Adolescence Neuropsychiatry Unit, Department of Neuroscience, Bambino Gesù Children’s Hospital, IRCCS, 00165 Rome, Italy; 3Department of Life Sciences and Public Health, Catholic University of the Sacred Heart, 00168 Rome, Italy; 4Systems Medicine Department, University of Rome Tor Vergata, 00133 Rome, Italy; 5Medical Direction, Bambino Gesù Children’s Hospital, IRCCS, 00165 Rome, Italy

**Keywords:** mental health disorders, adolescent, COVID-19 pandemic, emergency department, suicide, eating disorders, mood disorders

## Abstract

We described changes caused by the COVID-19 pandemic in the frequency of Emergency Department (ED) visits for mental health disorders (MHDs) in adolescents on a wider temporal range—that is, not just “the waves” of the pandemic—and characterized the profile of the adolescent seeking emergency psychiatric care. We conducted a retrospective longitudinal study by analyzing ED visits for MHDs from 10 March 2019 to 10 March 2021. A total of 1407 ED visits for MHDs were registered: 702 in the pre-COVID-19 and 707 in the COVID-19 period. The cumulative incidence of ED visits for MHDs was 1.22% in the pre-COVID-19 period and 1.77% in the COVID-19 period, with a statistically significant difference (*p* < 0.001). The principal characteristics of the adolescent with MHDs during the pandemic period: the odds of comorbidities decreased by 26% (*p* = 0.02), and the odds of transfer from other hospitals decreased by 71% (*p* < 0.001), while the odds of the ED presentation as first psychiatric episode were twice greater (*p* < 0.001). The risk of hospitalization increased by 54% (*p* = 0.001). Regarding psychopathology, the likelihood of attempted suicide increased by 74% during the pandemic (*p* = 0.02). The rate of mood and eating disorders grew significantly during the COVID-19 pandemic period (*p* = 0.005 and *p* = 0.031, respectively). Monitoring ED visits for MHDs and understanding changes in the profile of adolescents presenting to ED helps to reinforce the role of ED in identifying special clinical needs for these vulnerable patients in case of a future public health crisis.

## 1. Introduction

Besides the well-known negative effects on the physical health of individuals caused by the Severe Acute Respiratory Syndrome Coronavirus-2 (SARS-CoV-2), the COVID-19 pandemic represents a foreseeable and expected threat to mental health as well. In fact, high rates of psychiatric disorders in populations affected by natural disasters or past epidemics have been already reported globally across large-scale systematic reviews even prior to the COVID-19 outbreak [1,2,3]. Although adolescents are less susceptible to severe disease outcomes following SARS-CoV-2 infection than adults [4,5], they are not unscathed by the detrimental burden of the COVID-19 pandemic on mental health. Adolescence is a delicate phase of growth and development between childhood and adulthood, characterized by rapid biological, social, and physical changes, and psychological problems are known to emerge during this period [6]. The young are particularly exposed to the negative impact of events related to a pandemic because of a limited grasp of its intensity and inadequate coping strategies [7]. Lockdowns, social distancing, disruption of school and recreational routine, reduction of physical activity, loss of tutor time, excessive ‘screen time’ due to virtual learning and overuse of social or digital media, anxiety or fear of being infected and fear of infecting others, financial difficulties and familial instabilities, separation, and loss and grief due to separation from primary caregivers for illness, quarantine, or death are all major stressors affecting the mental health of children and adolescents [7,8,9]. In this context, the Emergency Department (ED) not only plays a leading role in mental health emergencies, as shown by growing pre-pandemic trends in mental health-related ED visits described in recent years in children and adolescents [10,11] but, increasingly, it is becoming a first point of contact for young people with acute or emerging mental health disorders (MHDs) [12], especially in case of difficulties or unavailability of the outpatient services.

Research highlighting the effects of the COVID-19 pandemic on ED visits for MHDs of adolescents is beginning to surface [13,14,15,16,17,18]. However, studies appraising the longitudinal psychological burden of the COVID-19 pandemic on adolescents in an emergency setting are not much [13,14], and this is important if we aim to “photograph” the effective COVID-19-related changes in the trend of the phenomenon. In addition, most studies focus only on psychopathology in ED, describing the overall frequency of adolescent psychiatric ED visits, and the frequency and trends in specific reasons for ED utilization [13,16,17,18,19]. However, we retain that it is also crucial to characterize the profile of the adolescent presenting to ED with MHDs during the COVID-19 pandemic because it provides the opportunity to reinforce the pivotal role of the ED in identifying and addressing mental health needs of a such vulnerable category of the patient such as the adolescent in case of future public health crises.

Therefore, the purpose of this study was, first, to describe changes caused by the COVID-19 pandemic in the frequency of ED visits for MHDs in adolescents on a wider temporal range that is not just “the waves” of the pandemic; second, we aimed to characterize the profile of the adolescent seeking emergency psychiatric care during the COVID-19 period.

## 2. Materials and Methods

We conducted a retrospective longitudinal study in the ED of the highest-volume tertiary children’s hospital in Rome and the entire Lazio region.

This study was approved by the Ethics Committee of our institution on 2 December 2020 (number: 2210_OPBG_2020) according to the Declaration of Helsinki (as revised in Seoul, Korea, October 2008). No informed consent was needed because of the retrospective and observational design of this study. All collected data were treated as strictly confidential and in anonymous form.

The study population included all ED visits for adolescents aged ≥ 12 and <18 years, from 10 March 2019 to 10 March 2021, presenting a MHD and receiving a final psychiatric diagnosis after a psychiatric consultation at ED. Patients were excluded if they left the hospital before receiving care because of the lack of a diagnosis and if there was missing data. Data were extracted from the electronic platform of the informatic health emergency system used in our region, and the identification of all patients was obtained by searching electronic health records both as free text and as International Classification of Diseases, 10th edition (ICD-10) diagnostic codes. Diagnoses have been considered according to the Diagnostic and Statistical Manual of Mental Disorder (DSM-5) criteria [20]. Under consideration of a senior pediatric neuropsychiatric consultant, we incorporated final diagnoses into ten categories: neurodevelopmental disorders, schizophrenia spectrum and other specific psychotic disorders, mood disorders, anxiety disorders, obsessive-compulsive and related disorders, trauma-related dissociation and dissociative disorders, eating disorders, disruptive, impulse control and conduct disorders, and “other disorders” (including substance-induced disorders, genetic syndrome, personality disorder, sleep disorder, specific learning disorders, conversion disorder, and gender dysphoria). We characterized the presence of aggressive behaviors as psychiatric epiphenomenon potentially associated with the ED presentation into five groups: self-harm, aggressive behaviors against others, suicidal ideation, attempted suicide, and psychomotor agitation. Referring to ED presentation, we also gathered demographic and clinical data, medical history, ED management and disposition after the pediatric and neuropsychiatric evaluation at ED.

We compared adolescent patients admitted to ED with MHDs before COVID-19 period (10 March 2019–10 March 2020) with patients admitted during COVID-19 period (11 March 2020–10 March 2021)

### Statistical Analysis

Statistical analysis was carried out through the software STATA/IC 14.2 version 2017. The normality of the distribution was verified by Skewness/Kurtosis test. Data were expressed as median values with an interquartile range (IQR), and direct comparisons were performed with Mann–Whitney rank-sum tests (as normality assumption was violated). Categorical outcomes were reported as percentages, and distributions of categorical data were compared with either Pearson’s χ2 test or Fisher’s exact test, as appropriate.

A multivariable logistic regression analysis was conducted to identify independent risk factors for ED admissions for MHDs in adolescents associated with COVID-19 period. We entered model variables considering the univariate significance. We described their point estimates (adjusted odds ratio [OR] for sex and age) together with their 95% confidence intervals (CIs). Statistical significance was set at *p* less than 0.05.

Moreover, we determined the cumulative incidence of visits for MHDs before the COVID-19 period and during the COVID-19 period, and we used the chi-squared test to determine whether there is a statistically significant difference between the two cumulative incidences. Finally, we calculated the percentage change of the cumulative incidence between the two periods.

## 3. Results

A total of 1407 ED visits for MHDs were registered during the overall study period, 702 in the pre-COVID-19 period and 707 in the COVID-19 period, respectively. The cumulative incidence of ED visits for MHDs was 1.22% [95% CI: 1.13 to 1.31] in the pre-COVID-19 period and 1.77% [95% CI: 1.64 to 1.91] in the pandemic period, with a statistically significant difference between the two rates (*p* < 0.001). Therefore, ED visits for MHDs increased by 45.08% in the COVID-19 period than the prior period (Table 1).

The demographic and clinical characteristics and the medical history and disposition of the total population and the two study groups are reported in Table 2.

Referring to the demographic features, the median age at ED admission was 15.75 years (14.5–16.83 years), but no significant difference was observed between the 2 groups as well as for the sex. On the other hand, other ethnicities (different from Italians) had more frequent ED evaluations in the pre-COVID-19 period (13.82% versus 7.93%; *p* < 0.001).

The rate of patients transferred from other hospitals was significantly lower in the COVID-19 period than in the pre-COVID-19 period (4.40%; % versus 9.83 *p* < 0.001).

Regarding the medical history, the time of symptom onset was significantly shorter in the COVID-19 period than in the prior period [1.5 days (1–7) versus 1 day (1–2); *p* < 0.001). The occurrence of comorbidity was significantly reduced during the COVID-19 period (26.95% versus 34.19%; *p* = 0.003). During the pandemic period, ED visits for MHDs were more frequent without prior mental health contact; therefore, adolescents at the “first contact” ED visit were 40.14% versus 25.50% during the pre-pandemic period (*p* < 0.001).

Referring to the management of patients with MHDs, the use of a secure room/security guard system at the ED was significantly more common during the COVID-19 period than the pre-COVID-19 period (21.56% versus 15.95%; *p* = 0.007); however, no significant difference was described in the need for pharmacological interventions and the use of physical restraint at ED.

Analyzing the presence of aggressive behaviors as a psychiatric epiphenomenon potentially associated with the ED presentation, the aggressive behavior against others was significantly less frequent in the pandemic period (23.97% versus 30.48%; *p* = 0.006), as well as the psychomotor agitation (51.77% versus 65.24%; *p* < 0.001); conversely, the rate of the suicidal ideation and the attempted suicide occurred more significantly in the COVID-19 period, 31.49% versus 23.79% (*p* = 0.002) and 13.33% versus 6.70% (*p* < 0.001), respectively.

After the ED evaluation regarding the final medical disposition, patients with MHDs needed more frequent hospitalization during the pandemic than in the prior period (45.67% versus 34.90%; *p* < 0.001).

Regarding the final diagnosis, more than a third of the total population was diagnosed with disruptive, impulse control and conduct disorders (35.96%), followed by mood disorders (25.66%), eating disorders (12.30%), anxiety disorders (6.75%), schizophrenia spectrum and other psychotic disorders (5.61%), neurodevelopmental disorder (4.34%), trauma-related dissociation and the dissociative disorders (1.92%), and obsessive-compulsive and related disorders (0.92%), whereas the category “other disorders” accounted for 19.55%. Compared to the pre-COVID-19 period, during the pandemic, the frequency of mood and eating disorders increased significantly, from 22.36% to 28.94% (*p* = 0.005) and from 10.40% to 14.18% (*p* = 0.03), respectively. Conversely, the frequency of schizophrenia spectrum and other psychotic disorders decreased from 6.84% to 4.40% (*p* = 0.047), and anxiety decreased from 9.12% to 4.40% (*p* < 0.001).

### Multivariable Logistic Analysis

We further performed a multivariate logistic regression analysis to obtain the statistically significant independent determinants for ED visits for MHDs in adolescent people associated with the COVID-19 period (Table 3).

The model showed that Italians were more than twice at risk of ED admission for MHDs during the COVID-19 period compared with other ethnic groups (OR: 2.64; 95% CI 1.79 to 3.88; *p* < 0.001). Adolescents during the COVID-19 period were 71% less likely to be transferred from another hospital (OR: 0.29; 95% CI: 0.18 to 0.47; *p* < 0.001). The odds of ED admission for MHDs during the COVID-19 period decreased by 1% for each day of the time of onset of symptoms. However, the OR was close to 1; therefore, the status was irrelevant for practical purposes.

The odds of comorbidity decreased by 26% in the COVID-19 period (OR: 0.74; 95% CI 0.58–0.95; *p* = 0.02). In addition, the odds of the ED presentation as the first psychiatric episode were twice greater during the COVID-19 period (OR 2.14; 95% CI 1.67 to 2.74; *p* < 0.001).

The likelihood of attempted suicide increased by 74% in the COVID-19 period (OR: 1.74; 95% CI: 1.11 to 2.71 *p* = 0.02), whereas the likelihood of psychomotor agitation decreased by 30% in the same period (OR: 0.70; 95% CI: 0.53 to 0.92; *p* = 0.01). Among final diagnosis, the logistic model did not identify any risk factor associated with ED admission for MHDs in the COVID-19 period; however, the risk of anxiety disorders decreased by 53% during the COVID-19 period (OR: 0.47; 95% CI: 0.28 to 0.77; *p* = 0.003).

The risk of hospitalization increased by 54% during the COVID-19 period (OR: 1.55; 95% CI: 1.19 to 2.03; *p* = 0.001).

## 4. Discussion

The trend of ED presentations for psychological distress in young people appears to have increased in recent years [21,22], and the COVID-19 pandemic has created a new social and health scenario impacting this trend as well. From this perspective, the strength of this study consists of longitudinally evaluating the psychological discomfort of the COVID-19 pandemic-related effects on mental health in a large sample of adolescents. This is achieved not only by describing changes in the frequency of ED visits for MHDs on a wider temporal range that is non-just “the waves” of the pandemic but also by characterizing the profile of the adolescent seeking emergency psychiatric care during the COVID-19 period.

In this current study, we registered a significant increase of 45.8% in the cumulative incidence of ED visits for MHDs in adolescents during the COVID-19 period. This finding was similar to other studies [13,14,23,24,25]. Nevertheless, two other studies reported a decline in psychiatric ED referrals during the pandemic [18,26]; however, both were limited to a short-term period of analysis near the lockdown. During the initial part of the lockdown, young people might have experienced a transient beneficial relief from some school-related stressors, such as social anxiety, school refusal, stressful educational requests from teachers and high expectations from parents, academic difficulties and tension surrounding social relationships and academic performance (i.e., plenty of homework, stressful assignments and tests, unhealthy competition between classmates), and bullying victimization [27,28]. The effects of the COVID-19 pandemic stressors on the MHDs probably became evident only months after their appearance.

Despite this rise in the incidence of MHDs, patients during the COVID-19 period were less likely to be transferred from other hospitals to our regional referral center for psychiatric emergencies, but this might be due to the fact that the COVID-19 pandemic has stretched the limits of health systems, including transport from other hospitals. Therefore, considering the collapse of the medical transport system, many parents/caregivers might have preferred to admit their children, on their own, to the referral pediatric hospital for MHDs.

We found that MHDs during the COVID-19 period were less likely to be associated with comorbidity. In the literature, some studies suggest that children and adolescents with mental illness are at risk of concurrent somatic/chronic illness and physical health problems, including asthma, eczema, gastrointestinal disorders, obesity, diabetes type 1, and headaches [29,30]. These results might be explained by the fact that the likelihood of the ED presentation as the first psychiatric episode was twice greater during the COVID-19 period than in the pre-COVID-19 period when patients more frequently reported a history of previous psychiatric episodes. In this case, patients with a known history of mental problems were probably already referred to pediatric psychiatric care centers where they could be intercepted for some clinical illness and readily referred to medical units for further investigations when needed. Therefore, they might be diagnosed with various comorbidities more frequently during the pre-COVID-19 period.

As mentioned above, another interesting aspect of the ED presentation during the COVID-19 period was the first contact ED visit for MHDs, i.e., the significant rate of previously healthy youths who were at a new onset of mental health problems in the emergency care setting. On this topic of “acute onset” of mental health problems in previously healthy individuals at the ED, some studies have reported up to half of the children and young people, who present to the ED, have no prior history or contact with the medical health care system; consequently, the ED has become the primary portal of access to the mental health system for young people not only before but also during the COVID-19 pandemic [13,14,18,23,24,25,26]. Nevertheless, the highest rate of ED presentations with an acute onset of MHDs during the COVID-19 period might be explained by the interaction between bio-psychological vulnerability and environmental stressors [31,32,33]. Particularly, crisis events can cause psychological distress in vulnerable groups such as adolescents [34,35]. In this sense, pandemics are described to be precursors to the mental health decline characterized by growing emotional stress, feelings of helplessness, and fear, which can escalate into mental illnesses, such as anxiety, depression, and post-traumatic stress symptoms in children and adolescents [7,36,37]. This concept might be also valid for the COVID-19 pandemic. Psychological-social stressors related to the COVID-19 pandemic to which adolescents have been exposed are multiple: home quarantine, closure of schools, and interruption in sports, cultural activities and any kind of public or private event that are all factors that have determined a loss of social connectivity and a drastic routine disruption. In addition, a decline in physical activity, loss of tutor time, and excessive ‘screen time’ due to virtual learning and overuse of social or digital media bombarded with negative news can adversely affect mental health as well as the separation, loss, and trouble caused by the separation from primary caregivers because of illness, lockdown, or death or the anxiety or terror for their own or their parents’ death [6,7,38,39,40,41]. Altogether, these multiple stressors might work as an “explosive mixture” and might abruptly and acutely destroy the personal and social balance of the adolescent who basically suffers a lack of positive coping skills and resilience during periods of adjustment [40,41].

Referring to psychiatric epiphenomena, our data demonstrated that suicide attempts constituted a risk factor for ED presentation during the COVID-19 period. This finding coincided with the results of most studies [42,43,44,45] and draws alert attention to the “dual pandemic”: COVID-19 and suicide behaviors [45]. We did not analyze the temporal trend of suicidal phenomena; thus, we cannot see if there was some variation described in other studies. In fact, the literature reports a trend of suicidal behaviors characterized by an initial drop during the first wave of the pandemic followed by a persisting rise as the pandemic progressed [43,44,46,47], which resembles reports of people involved in catastrophic events. This trend is named the “honeymoon effect” [48], which describes a phase after a disaster dominated by a feeling of a short-lived sense of optimism, followed by the disillusionment phase where people are coming to grips with the reality of their situation [49]. In addition, school closures could have restrained the discomfort and anxiety due to the academic burden and social problems with peers in the early phases of the lockdown and consequently could have limited adolescents’ anxiety and depressive symptoms, at least initially, and might have controlled the potentially high risk of suicide [27,50]. However, other studies did not find this increasing tendency [13,14,51]. The reason for this discrepancy might rely on the fact that suicidal thoughts or behaviors may not come out until later in the pandemic and may not have occurred evenly among groups of youth [52]. Nevertheless, adolescence is characterized by a higher risk of developing suicidal phenomena since it is a “storm period” characterized by an imbalance between an amplified susceptibility to motivational strivings and an immature cognitive control which could weaken the impulse inhibition to suboptimal actions such as self-harm [53]. There were alerts that adolescent suicidal behaviors were on the rise over the last decade. The pandemic could have worsened the situation, increasing psychological unease among adolescents. Therefore, it is not surprising that in our study, adolescents admitted to ED for MHDs presented higher numbers of suicide attempts. The COVID-19 pandemic-related risk factors affecting suicidal tendencies are particularly represented by the quarantine and social distancing [54,55]. These circumstances determine the growth of negative emotions (anger, fear [56], pervasive sense of uncertainty for the future, loneliness, and unhappiness [57]) and correspond with reduced psychological well-being and the onset of psychological symptoms and emotional disorders (depression, anxiety, insomnia, post-traumatic symptoms [58], and eating disorders [59]). Furthermore, because of school closures, the use of virtual learning has become a routine which could have been a source of stress, frustration, and disadvantage for some adolescents [60], especially from families of lower socio-economic status who are unable to attend distance learning sessions for the lack of resources such as the internet. Other factors are connected to the suicidal risk phenomena: the excessive use of social platforms that increases isolation and, in turn, the risk of suicide [61]. In fact, sharing and posting material reporting, pushing for, normalizing, or supporting others to join in dangerous activities may lead to serious injury or death, provide fodder for adolescent imitative behavior, and therefore influence suicidal behavior [62]. Moreover, financial crises in families [63] and fear of contagion [63] can have traumatic and enduring detrimental effects on adolescents. Likewise, grief following the death of relatives or family members has also been recognized as a risk factor for suicide in adolescence; during the first year of the pandemic, several reports tell us of mothers, fathers, and grandparents who have died from COVID-19 [64]. In addition, the higher risk of exposure to neglect, physical, emotional, and sexual abuse and violence exacerbated by home confinement might have led to increased suicide rates among adolescents [63].

Among other psychiatric epiphenomena, aggression against others and psychomotor agitation were significantly associated with the pre-COVID-19 period. This finding might seem a paradox; in fact, most studies show there has been an increase in aggressive behavior during the lockdown, not only in adults but also in youths, especially towards parents [64]. Nevertheless, a Chinese study carried out among students during the COVID-19 crisis showed that the pandemic was not significantly related to aggressiveness, which, on the contrary, appeared to have been reduced in the worst-hit geographical zones [41]. This is true for Italy, which has been the first among European countries to suffer severe effects of the pandemic.

Analyzing the final diagnosis related-ED presentations, the mood disorders were significantly associated with the COVID-19 period, although, according to the logistic model, this diagnosis did not result to be a predictive factor of ED visits for MHDs during the COVID-19 period [41,65,66,67,68]. However, the study conducted by Chen et al. did not find a significant correlation between depression and COVID-19 [69].

Paradoxically, in our study, anxiety disorders were a predictive factor of ED visits for MHDs during the pre-COVID-19 period compared with the COVID-19 period. Instead, most studies identified an association between anxiety and the COVID-19 pandemic [41,66,67,70]. Indeed, other studies described no change in internalizing symptoms [71] or a reduction in risk for anxiety [72] or no significant correlation between COVID-19 and anxiety among adolescents [69]. In another study dealing with adolescent depression and anxiety symptoms, elaborating on data from 12 longitudinal studies, depressive symptoms increased significantly, but anxiety symptoms did not [73]. There are several explanations for this discrepancy. First, the methodology of the study is detrimental. In fact, the use of broad screening questionnaires adopted in the majority of the above-mentioned cross-sectional studies might have limited the possibility of observed changes in specific forms of anxiety [73]. Moreover, the use of a variety of anxiety scales among adolescents might have influenced different results [73]. Second, anxiety has polyhedric declinations. In fact, some forms of anxiety might increase during the pandemic while other forms might decrease [73]. For example, social anxiety may have been momentarily mitigated as a result of reduced opportunities for social connections and loosened social pressures, whereas generalized anxiety may have been boosted parallel to the global pandemic [73]. Third, anxiety may have varied during the pandemic based on the circumstances to which adolescents were exposed [73]. For example, in the questionnaires used in some studies, several standardized measures of anxiety comprise items that are difficultly suitable and applicable to new situations such as the local restrictions (e.g., “I get stomach aches at school”) and consequently may underreport experienced anxiety or camouflage the one that has no behavioral expression when adolescents are largely at home [73]. At the same time, recovery from school and social activities might lead to an increase in anxiety symptoms because of missed opportunities for exposure [73].

Another salient result of our study was the association between the increased rate of eating disorders and the COVID-19 period. This finding was supported by other previous studies demonstrating that the COVID-19 pandemic promoted the development and worsening of eating disorder symptoms in young people [74,75]. For at-risk youth, the mental and physical effects of disruptions in school, sports, cultural, social, and leisure activities sparked or worsened irregular eating behaviors. Moreover, the pandemic intensified the usage of social media, which has been associated with worsening symptoms in individuals presenting eating disorders; the consequent increase in online interactions with peers may increase self-criticism and negative appearance-related comparisons [75]. Eating disorders originate from various concatenated biological, psychological, and social factors upon which operate other triggering factors, such as particularly stressful situations. The lockdown has favored, specifically in teenagers, the establishment of several triggering factors, among which are social isolation, the uncertainty of academic prospects, devices and regulations of protection which has caused forced isolation between peers, and fear of infection often linked to a sense of powerlessness [68,76,77]. The cited circumstances might lead to an increase in eating restrictions or, on the contrary, frequent sessions of uncontrolled eating. This can result in a major focus on body image, giving space to a more prominent consideration of the body, consequently of food and exercise, and the rise of dysfunctional habits both in excess and deficit. [78]. As reported by Rogers et al., the changes caused by the COVID-19 pandemic might generate a fertile ground for the emergence or augmentation of eating disorders risk and symptoms, and, at the same time, it might undermine protective factors against eating disorders and might limit access to healthcare services [79]. Three pathways exist by which this pandemic may aggravate the risk of eating disorders [79]. Firstly, the compromised opportunity of physical outdoor activities owing to “stay-at-home” restrictions and disruptions to everyday life may increase weight and shape concerns and negatively impact eating, exercise, and sleeping patterns, which may, in turn, increase eating disorders risk and symptoms. Likewise, the pandemic and relative social distancing measures may compromise social support and adaptive coping skills, which are protective factors for psychological adaptation to distress and mental well-being. Depriving individuals of these protective factors might increase eating disorder risk and symptoms [79]. Secondly, there is evidence highlighting a positive association between prolonged media exposure (especially social media) and the development of eating disorders through the dissemination of content promoting harmful concepts of body image and unhealthy eating behaviors. In addition, the enlarged use of videoconferences as a needed alternative to in-person school due to stay-at-home orders may also increase the risk for eating disorders by increasing preoccupation with appearance [79]. Lastly, fears of contagion may increase eating disorder symptoms, specifically related to health concerns or the pursuit of restrictive diets focused on increasing immunity [79]. Elevated rates of stress due to the pandemic and social isolation may also contribute to increased risk [79]. Altered food accessibility, food insecurity due to the global economic crisis caused by COVID-19, and barrier to health care during the pandemic may contribute to rising rates [75].

Finally, another data coming to light in our study was the higher rate of mental health-related hospitalizations among adolescents. This result was also confirmed nationally by data from the Italian Pediatric Society (SIP) [80] and was like other studies [81,82]. Nevertheless, others reported a significant reduction in hospitalizations [13,14,18]. This discrepancy might rely on different reasons. For example, in the case of the study setting during the early phases of the COVID-19 pandemic, especially during the weeks of COVID-19-induced social lockdown [18] (i.e., the research by Davico et al.), the decision to hospitalize patients did not rely not on the severity of psychiatric illness, but also on purpose to avoid the exposure of patients to the risk of COVID-19 infection in the wards and to address hospital resources towards the most complicated patients [26]. It is also possible that although there was a larger need to seek mental health care during the initial phases of the pandemic, some countries might have established standards of telehealth psychiatric care for children and adolescents, and the telehealth services were largely sufficient and effective in fulfilling the needs of patients. In fact, Sánchez-Guarnido et al. [83] reported that for patients who received teletherapy during the lockdown, ED visits and hospitalization rates were reduced four and six months after the first wave of COVID-19. Using teletherapy during stressful periods, when in-person sessions are not feasible, was a protective factor against hospitalization, particularly in the medium term [83]. Moreover, to minimize the risk of exposure of hospitalized patients to the COVID-19 infection, visits from family and friends to patients were banned or severely restricted. These strict visitation policies may have caused patients and their parents to be more diffident and even refuse hospitalization compared to pre-pandemic periods [14]. In addition, the decision of the clinician to hospitalize or discharge the patient was typically and mainly based on the severity of conditions [18]; thus, the increased rate of adolescents requiring psychiatric hospitalization described in our study might be due to the increased severity of mental illness. This hypothesis was supported by Millner et al., who suggested youth hospitalized during the pandemic reported increased severity of mental health symptoms compared to those hospitalized before the pandemic [84].

Our study is characterized by certain limitations. First, our results are based on a single, large, and tertiary pediatric center and might not be generalizable to other clinical settings. However, our institution is recognized as the referral center for the Lazio region for pediatric psychiatric emergencies, and it has the largest number of ED visits in Rome and the Lazio region, accounting for an annual average of 5200 outpatient visits, 2700 day hospital admissions, and 240 hospital admissions. Second, it cannot exclude that the decision to hospitalize the patient after the ED visit, although based on the severity of psychiatric illness, might be biased by the greater availability of hospital beds. Another limitation is the retrospective design of this study and, therefore, limits our ability to gather more data about patients, such as information regarding patient complaints, which could be relevant to better understanding the reasons for the results shown in this study. Finally, the temporal factor is certainly a crucial aspect to consider regarding results because every individual manifests time-based behavioral and emotional responses to an event crisis such as a pandemic or natural disaster. As suggested by Lin YH et al. [85], to avoid bias selection, a longitudinal design or high temporal resolution would help provide a strong methodologic basis for investigating pandemic-related mental health impacts. Therefore, this limitation, partly based on the temporal factor, is overtaken thanks to the methodology. However, a bias might be represented by the fact that we did not analyze the temporal trend of mental health disorders (MHDs) and thus cannot evaluate if there was some variation as described in other studies.

## 5. Conclusions

Our findings not only showed a striking increase in the rate of mental health ED visits among adolescents during the first year of the pandemic but suggested that the pandemic has resulted in a significant change in the profile of the young patient who visited ED for MHDs. In summary, the principal characteristics include a lower frequency of associated comorbidities and transfer by another hospital and a greater frequency of the first-time, acute episode of ED presentation and hospital admission for MHDs. Moreover, regarding the specific psychiatric picture of these patients, suicide attempt was identified as a risk factor for ED presentation during the COVID-19 pandemic; the rate of mood and eating disorders grew significantly during the COVID-19 period, while the rate of anxiety disorders rose before the pandemic.

A longitudinal evaluation of the impact of the pandemic on the frequency of ED visits for MHDs among adolescents and an understanding of changes in characteristics of patients seeking emergency mental care may reinforce the pivotal role of ED in the assessment, treatment, and coordination care of a fragile population during a future public health crisis.

## Figures and Tables

**Table 1 children-10-00578-t001:** The frequency of ED visits for MHDs among adolescents, and the change in the frequency before and during the COVID-19 period.

	Pre-COVID-19 Group	COVID-19 Group	*p* Value
Number of ED visits for MHDs	702	707	<0.001
Number of total ED visits	57,384	39,832	
The cumulative incidence of ED visits for MHDs disorders per 100 children [95% CI]	1.22% [1.13 to 1.31]	1.77% [1.64 to 1.91]	
Percentage change of the cumulative incidence of ED visits for MHDs	+45.08% increase		

**Table 2 children-10-00578-t002:** Characteristics of mental health emergency visits among adolescents before and during the COVID-19 pandemic.

Characteristics	Total Population n = 1407	Pre-COVID-19 Period n = 702	COVID-19 Period n = 707	*p* Value
Sex, n (%)				0.50
Female	543 (38.59)	277 (39.46)	266 (37.73)	
Male	864 (61.41)	425 (60.54)	439 (62.27)	
Age (years), median (IQR)	189 (174–202) 15.75 (14.5–16.83)	189 (173–200) 15.75 (14.42–12.66)	189 (175–204) 15.75 (14.58–17)	0.1
Ethnicity, n (%)				
Italian	1259 (89.48)	605 (86.18)	654 (92.77)	<0.001
Other ethnicities	148 (10.52)	97 (13.82)	51 (7.23)	
Arrival to hospital by ambulance, n (%)	566 (40.23)	274 (39.03)	292 (41.42)	0.36
Transferred by another hospital, n (%)	100 (7.11)	69 (9.83)	31 (4.40)	<0.001
Time of symptom onset, (days), median (IQR)	1 (1–3)	1.5 (1–7)	1 (1–2)	<0.001
Comorbidity, n (%)	430 (30.56)	240 (34.19)	190 (26.95)	0.003
Onset of MHDs, n (%)				
“First contact” ED visits	462 (32.84)	179 (25.50)	283 (40.14)	<0.001
No previous psychiatric history	945 (67.16)	523 (74.50)	422 (59.86)	
Intentional exposure to toxic substances, n (%)	199 (14.14)	100 (14.25)	99 (14.04)	0.91
On current medical treatment, n (%)	752 (53.45)	375 (53.42)	377 (53.48)	0.98
Need for pharmacologic interventions at ED, n (%)	74 (5.26)	34 (4.84)	40 (5.67)	0.48
Need for the use of physical restraint at ED, n (%)	12(0.85)	9 (1.28)	3 (0.43)	0.07
Use of our secure room/security guard system at ED, n (%)	264 (18.76)	112 (15.95)	152 (21.56)	0.007
Positive toxicologic screening at ED, n (%)	113 (8.03)	55 (7.83)	58 (8.23)	0.78
Need for specialist consultations in addition to psychiatric consultation at ED, n (%)	81 (5.76)	39 (5.56)	42 (5.96)	0.75
Need for hospitalization, n (%)	567 (40.30)	245 (34.90)	322 (45.67)	<0.001
Aggressive behavior, n (%)				
Self-harm	339 (24.09)	162 (23.08)	177 (25.11)	0.37
Aggressive behavior against others	383 (27.22)	214 (30.48)	169 (23.97)	0.006
Suicidal ideation	389 (27.65)	167 (23.79)	222 (31.49)	0.001
Attempted suicide	141 (10.02)	47 (6.70)	94 (13.33)	<0.001
Psychomotor agitation	823 (58.49)	458 (65.24)	365 (51.77)	<0.001
Final diagnosis, n (%)				
Neurodevelopmental disorder	61 (4.34)	33 (4.70)	28 (3.97)	0.50
Schizophrenia spectrum and other psychotic disorders	79 (5.61)	48 (6.84)	31 (4.40)	0.047
Mood disorders	361 (25.66)	157 (22.36)	204 (28.94)	0.005
Anxiety disorders	95 (6.75)	64 (9.12)	31 (4.40)	<0.001
Obsessive-compulsive and related disorders	13 (0.92)	9 (1.28)	4 (0.57)	0.16
Trauma-related dissociation and the dissociative disorders	27 (1.92)	13 (1.85)	14 (1.99)	0.85
Eating disorders	173 (12.30)	73 (10.40)	100 (14.18)	0.031
Disruptive, impulse control and conduct disorders	506 (35.96)	256 (36.47)	250 (35.46)	0.70
Other disorders	275 (19.55)	132 (18.80)	143 (20.28)	0.48

**Table 3 children-10-00578-t003:** Multivariable logistic regression analysis exploring risk factors associated with ED visits for MHDs in adolescents during the COVID-19 pandemic.

COVID-19 Period	OR	Std. Err.	z	*p* > |z|	95% CI	
Sex (female versus male)	1.15	0.14	1.13	0.26	0.90	1.47
Age	1.01	0.003	1.88	0.06	1.00	1.01
Ethnicity (Italians versus others)	2.64	0.52	4.93	<0.001	1.79	3.88
Comorbidity	0.74	0.09	−2.34	0.02	0.58	0.95
Transferred by another hospital	0.29	0.07	−5.08	<0.001	0.18	0.47
Time of symptom onset	0.99	0.001	−2.13	0.03	0.994	1.00
Onset of MHDs (“First contact” ED visit versus no previous psychiatric history)	2.14	0.27	6.07	<0.001	1.67	2.74
Use of our secure room/security guard system at ED	0.93	0.18	0.38	0.70	0.64	1.35
Self-harm	0.87	0.13	−0.95	0.34	0.64	1.16
Suicidal Ideation	1.08	0.19	0.42	0.68	0.76	1.52
Attempted suicide	1.74	0.40	2.43	0.015	1.11	2.71
Psychomotor agitation	0.70	0.10	−2.57	0.01	0.53	0.92
Need for hospitalization	1.55	0.21	3.22	0.001	1.19	2.03
Schizophrenia Spectrum and Other Psychotic Disorders	0.63	0.17	−1.75	0.08	0.37	1.06
Mood Disorders	1.02	0.16	0.14	0.88	0.75	1.40
Anxiety Disorders	0.47	0.12	−3.02	0.003	0.28	0.77
Eating Disorders	0.93	0.21	−0.33	0.74	0.60	1.43
Constant	0.31	0.19	−1.91	0.06	0.09	1.03

OR—odds ratio; Std. Err.—standard errors associated with the OR; z and *p* > |z|—these columns provide the z-value and 2-tailed *p*-value used in testing the null hypothesis that the coefficient (parameter) is 0; 95% CI—this shows a 95% confidence interval.

## Data Availability

Data presented in this study are available on request from the corresponding author.

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
