# Peer review of "Has Anything Changed in the Frequency of Emergency Department Visits and the Profile of the Adolescent Seeking Emergency Mental Care during the COVID-19 Pandemic?"

_children, 2023, doi:10.3390/children10030578_

Round 1
Reviewer 1 Report
Dear Authors,
I provide a review of the manuscript in the pdf attachment.
Kind regards.

Author Response
We added the title of the table 3 as suggested
Reviewer 2 Report
The followings are my comments:
1. In the introduction section, the authors emphasized that the samples were collected in a wider temporal range rather than the "waves" of the pandemic. However, in a natural disaster, the temporal factor may significantly affect the disaster's damage to human beings, especially their behaviors. Bias may be caused if the confounding effect of the temporal factor is not considered during the model’s establishment.
2. In the materials and methods section, evidence or references should be provided for the classifications of the patients' final diagnoses and their aggressive behaviors.
3. In table 2, the number "48 " in the variable "other ethnicities " cell needs to be corrected.
4. In the results section, the description of the statistical result of the variable “self-harm” was completely different to the data of the same variable in Table 2 (p=0.006 vs p=0.37, respectively).
5. At this point, I suggested the authors should check all data thoroughly to confirm the data correctness in this manuscript before further review.
Author Response
1.In the introduction section, the authors emphasized that the samples were collected in a wider temporal range rather than the "waves" of the pandemic. However, in a natural disaster, the temporal factor may significantly affect the disaster's damage to human beings, especially their behaviors. Bias may be caused if the confounding effect of the temporal factor is not considered during the model’s establishment.
I thank you for the annotation. The temporal factor is certainly a crucial aspect to consider discussing results because every individual manifests time-based behavioral and emotional responses to an event crisis such as a pandemic or a natural disaster. As suggested by Lin YH et al [1], to avoid bias selection a longitudinal design or high temporal resolution would help provide a strong methodologic basis for investigating pandemic related mental health impacts. Therefore, in partly this limitation based on temporal factor is overtaken thanks to the methodology. However, a bias might be represented by the fact that we did not analyze in our study the temporal trend of the mental health disorders (MHDs) so cannot see if there was some variation as described in other study. We added this bias in the discussion in the part specifying limitations of study.
1Lin YH, Chen CY, Wu SI. Efficiency and quality of data collection among public mental health surveys conducted during the COVID-19 pandemic: Systematic review. Journal of Medical Internet Research. 2021;23(2).
2.In the materials and methods section, evidence or references should be provided for the classifications of the patients' final diagnoses and their aggressive behaviors.
We added references in the manuscript, as requested, concerning the classification of the “final diagnoses” that was derived from Diagnostic and Statistical Manual of Mental Disorders Fifth Edition Text Revision (DSM-V-TR). While the classification of aggressive behaviors was based on medical judgement of a senior pediatric neuropsychiatric consultant considering aggressive behaviors as epiphenomenon of a mental health disorders.
- In table 2, the number "48 " in the variable "other ethnicities " cell needs to be corrected.
We modified the typing error (absolute number of the variable “Other ethnicities”-: 148 instead of 48.
- In the results section, the description of the statistical result of the variable “self-harm” was completely different to the data of the same variable in Table 2 (p=0.006 vs p=0.37, respectively).
We intended the variable “aggressive behavior against others” instead “self-harm”. It was a typing error, so we corrected it in the text concerning results section. Results regarding the mentioned variable are right, as well as comments on this variable in discussion section.
- At this point, I suggested the authors should check all data thoroughly to confirm the data correctness in this manuscript before further review.
We verified all results as requested.